# Microstrip-Fed 3D-Printed H-Sectorial Horn Phased Array

**DOI:** 10.3390/s22145329

**Published:** 2022-07-16

**Authors:** Ivan Zhou, Lluís Pradell, José Maria Villegas, Neus Vidal, Miquel Albert, Lluís Jofre, Jordi Romeu

**Affiliations:** 1School of Telecommunication Engineering, Universitat Politècnica de Catalunya, 08034 Barcelona, Spain; lluis.pradell@upc.edu (L.P.); miquel.albert@estudiantat.upc.edu (M.A.); luis.jofre@upc.edu (L.J.); jordi.romeu-robert@upc.edu (J.R.); 2Department of Electronics and Biomedical Engineering, Universitat de Barcelona, 08028 Barcelona, Spain; j.m.lopez_villegas@ub.edu (J.M.V.); nvidal@ub.edu (N.V.)

**Keywords:** printed antennas, 3D antennas, horns, low-loss antennas, 5G millimeter-wave antennas

## Abstract

A 3D-printed phased array consisting of four H-Sectorial horn antennas of 200 g weight with an ultra-wideband rectangular-waveguide-to-microstrip-line transition operating over the whole LMDS and K bands (24.25–29.5 GHz) is presented. The transition is based on exciting three overlapped transversal patches that radiate into the waveguide. The transition provides very low insertion losses, ranging from 0.30 dB to 0.67 dB over the whole band of operation (23.5–30.4 GHz). The measured fractional bandwidth of the phased array including the transition was 20.8% (24.75–30.3 GHz). The antenna was measured for six different scanning angles corresponding to six different progressive phases α, ranging from 0° to 140° at the central frequency band of operation of 26.5 GHz. The maximum gain was found in the broadside direction α = 0°, with 15.2 dB and efficiency η = 78.5%, while the minimum was found for α = 140°, with 13.7 dB and η = 91.2%.

## 1. Introduction

Fifth generation (5G) millimeter-wave (mmWave) communication is a promising solution to the problem of the demand on network capacity, providing low latency and high data speed. However, higher propagation losses will also be introduced, requiring beamforming capabilities [1] for the transmitters in order to mitigate these effects [2].

The availability of chip beamformers [3] at these frequencies makes microstrip line (ML)-based circuitry the optimum solution for the implementation of RF electronics. This is not the best choice for antenna design, due to the propagation losses inside the substrate of an ML. Investigations in [4,5] revealed that the ML is more suitable for feeding arrays of a small or medium size, because the existence of the inevitable dielectric loss in substrates limits the antenna gain. It is therefore necessary to have an alternative technology for designing high-gain antennas that can easily be integrated into beamforming chips.

In this regard, horn antennas are proposed, where the gain can be increased by enlarging the radiating aperture and the costs and weights can be reduced by using additive manufacturing techniques such as 3D printing [6]. There are multiple printing methods, such as selective laser sintering (SLS) [7], where a laser selectively sinters the particles of a polymer powder, fusing them together and building a part layer by layer; fused deposition modeling (FDM) [8], where molten plastic is extruded from a computer-controlled hot-end and cooled to form a part; stereolithography (SLA) [9], where a light source is used to selectively harden photo-activated resins; material jetting (MJ) [10], where the printheads are used to deposit a liquid photoreactive material onto a build platform layer upon layer; and direct metal laser sintering (DMLS) [11], similar to FDM but with a metallic powder. In [12], a fully 3D-printed complex corporate feeding network with 256 horns was successfully manufactured using DMLS, but the cost was high. Many cost-effective horn antenna designs have been produced using FDM [13,14,15], but although they represent a large variety of applications and a successful manufacturing process, transitions from the horn to the ML (equivalent to RW–ML transitions) are usually required to provide an effective connection with the RF chipset, and this has not yet been reported in any research.

In this context, a backshort-less ML-to-RW transition via proximity coupling through a patch antenna was proposed [16,17]. Although it offers a good fractional bandwidth (FBW) of 18% by controlling the via holes, it is fed from the broad wall of the RW, which is not convenient because compact $\lambda_{g}/2$-spaced arrays cannot be implemented. Transitions from the narrow wall of the RW to the ML are advantageous, due to the compactness in this transversal direction with respect to the excitation port. In [18], the RW is excited through a transversal patch antenna. However, intrusion elements inside the RW must be inserted in order to enhance the bandwidth from 11% to 15%, increasing the fabrication complexity. The need for intrusion elements is avoided in [19] by using a V-shaped aperture-coupled patch. Although this avoids the implementation of a back-shorting cavity, only a 7.5% FBW is achieved. In [20], the FBW is increased up to 11% through a patch fed by a coupled ML. In comparison with the transitions presented in this paper [16,18,19,20], a top–side narrow-wall ML-to-RW transition is designed by using three overlapped transversal patches, offering very low insertion loss (IL) (maximum peak of 0.67 dB) and a very wide FBW (22.5%), covering the whole 23.5–30.4 GHz range for 5G mmWave LMDS and K bands. The presented transition is integrated into (and used to validate) a 3D-printed phased array (PA) using the MJ technique of H-sectorial horn antennas, with a novel corrugated layer that reduces the blind scanning problem and a simple metallization procedure consisting of only covering the outside part of the horns, making the PA ready to be integrated with any RF chipset. This has not been reported in any previous research. The measured results prove the feasibility of the 3D-printing technology for manufacturing complex active antenna systems such as PAs.

## 2. Transition Design

The focus of this first section is on the conversion of the Q-TEM mode of a 50 Ω ML to the $TE_{10}$ mode of a WR-34 (with dimensions $W_{rw}$ × $L_{rw}$) located on the same side of the substrate as the ML, so that it can later be used to feed an array of four $\lambda_{g}/2$-spaced H-sectorial horn antennas using an ML. Figure 1 shows the top view of the transition, Figure 2 shows its isometric view and Table 1 shows the dimensions of the transition. The transition is designed using a 0.81 mm thick RO4003 substrate of Er = 3.55 and tan δ = 0.0027. A full-wave electromagnetic simulation tool, CST Microwave Studio, was used for its optimization. In order to excite the $TE_{10}$ mode from the RW, the excitation of transversal currents in the $\hat{x}$ direction is required. An array of three overlapped transversal patches, as proposed in [21], was used. The length Lp of the patches, the width $W_{p}$, the inter-element distance $d_{y}$ and the array position with respect to the entrance of the RW $y_{0}$ were jointly optimized to provide maximum bandwidth while maintaining minimum IL. Re-optimization of the transition was carried out when integrating it with the 3D-printed PA shown in the next section. The overlapping of these patches of length $0.2\lambda_{g}$ is crucial for the excitation of transversal $J\hat{x}$ currents. The array is fed by an ML of width $W_{f}$ and is matched to a 50 Ω circuit with a stepped section of dimensions $W_{t}$ × $L_{t}$. The input to the RW has dimensions Win × Hin, and ideally these should be as small as possible to avoid leakage from the RW, while also preventing short-circuiting with the ML. The final dimension was chosen considering the manufacturing feasibility.

The narrow wall of the RW is stepped (see Figure 2) from dimensions Wb × Lrw × Ha to Wrw × Lrw × Hb, so that the dimensions of the H-sectorial horn antenna along the $\hat{x}$ direction can be reduced for the integration of a mutual coupling reduction (MCR) layer, which is introduced in Section 3. In Figure 3, we show the simulated S-parameters of the ML–RW transition. We can observe a good −10 dB input reflection coefficient ranging from 23.5 GHz to 30.4 GHz. Note that this transition was optimized jointly with the 3D-printed PA above it.

## 3. Phased Array Design

In Figure 4, an exploded view of the whole microstrip-fed 3D-printed PA of four active H-Sectorial horns is shown. The antenna was designed to be attached through four 2.4 mm coaxial cables to the F5288 digital beamformer from RENESAS, achieving four active channels. It is composed of five different layers, L0,L1,L2,L3 and L4. The first layer L0 consists of four 2.4 mm female connectors attached to the ML printed on the RO4003C substrate, which are used to feed the ML–RW transition presented in Section 2. The distance between the connectors and the transition is 25 mm. Layer *L*1 is a horn coupler that attaches layer *L*0 to layer *L*2. In turn, *L*2 is a support for six H-sectorial horns, implemented on layer *L*3 through two M2.5 screwed holes. Although a direct chip-to-ML integration would be a more elegant option to reduce design complexity and bulkiness, as well as cable losses, this was not the main goal of this research. The main goal was to demonstrate the feasibility of 3D-printing technology for manufacturing complex antenna systems such as PAs.

Only four horns are active, with two lateral dummies placed on the sides of the phased array in order to symmetrize the radiation pattern. The top L4 layer consists of a periodic arrangement of nails to reduce the mutual coupling (MC) between each horn antenna. Thus, the blind scanning issue is minimized, maintaining a −10 dB matching of all the four active horns for a wide range of scanning angles related to a progressive phase, ranging from 0° to 140°. The support, L1,L2,L3 and L4 were fully 3D-printed with an Objet Connex 1 printer from Stratasys, using the material jetting technique, with a tolerance of ±0.05 mm on the Z axis and ±0.1 mm on the X and Y axes. The photosensitive resin ink was High Temperature RGD525, with an electrical permittivity of $E_{r}$ = 2.95 and a loss tangent of 0.0175. All the printed materials except L2 were metallized using copper electrodeposition.

### 3.1. Horn Antenna

Six H-sectorial horns were designed and 3D-printed, with dimensions as shown in Figure 5. The horns had height $L_{h}$, width $W_{h}$ and thickness $W_{y}$. The entrance of the horn had width $X_{W}$ and height $X_{h}$, which was optimized in order to maximize the bandwidth of the joint horn and transition design. The inner side of the horn was empty in order to reduce the transmission losses of the EM fields, and the thickness of the walls of each horn was a constant of value $W_{wall}$.

Only the outer lateral parts of the horn were metallized using the conventional electrodeposition technique, rather than metallizing both outer and inner parts as is usually the case, increasing the reliability of a complete surface copper deposition process. An optimum metallization setup would be a plastic part whose surfaces are almost parallel to the fields created by the electrodes, so that the particles of copper travel directly to the surfaces and attach to them. This is the case when metallizing the outer part of the horn. However, when metallizing the inner parts of the horn (which was avoided in our case), the deposition of the copper is complicated by the presence of surfaces perpendicular to these fields.

### 3.2. Mutual Coupling Reduction

A mutual coupling reduction layer L4 was designed in order to reduce the well-known blind scanning problem [22] of phased arrays. This layer used periodic nails $p_{ycorr}$ of a trapezoidal shape with a squared bottom shape $W_{Anail}$ × $W_{Anail}$, top shape $W_{Bnail}$ × $W_{Bnail}$ and height $H_{nail}$, see Figure 6. The periodic nails, also known as high-impedance surfaces, are a preferred option compared to $\lambda_{g}/4$ slits, due to the intrinsic wide-band behaviour.

In order to find the optimal parameters to reduce the blind scanning problem, the MC between adjacent horns must be reduced. By selecting only two horns, each one with a simulation port, it is possible to optimize their dimensions by finding the minimum coupling between them. In Figure 7, we can see a comparison of the input reflection coefficients (a) without the nails and (b) with nails. For each port of the horn, there are six traces relating to the progressive phase from 0° to 140° at 26.5 GHz. The reason for showing both plots in this compact way is to show that all the ports from the PA are well matched for every scanning angle when using the mutual coupling reduction layer. We can clearly see that port 4 becomes mismatched when a progressive phase along the arrays differs from 0, which will clearly reduce the gain of the beamformer.

## 4. Results

This section discusses the experimental results for the designed phased array. Figure 8 shows the manufactured and mounted PA connected to the beamformer, Figure 9 shows the results of the input reflection coefficients, Figure 10 shows the scanning pattern only in the azimuthal plane of the PA in dB and Figure 11 shows the phased array gain with respect to the frequency for three different progressive phases. In Figure 9, the input reflection coefficients correspond to each individual port (when measuring one port the rest remain unexcited). Note that measuring the input reflection coefficient of each port when exciting all ports and using the beamformer for scanning in the E-plane was not possible with this setup. The measured input reflection coefficients are all slightly shifted to higher frequency, with a wider bandwidth behaviour and an unexpected resonance at 31.25 GHz. The antenna is wideband in nature as the horn is a non-resonant structure, offering a measured FBW of 20.8%.

Only the radiation pattern in the E-plane was measured for each scanning angle, using the digital beamformer at four different frequencies ranging only from 26.5 GHz to 29.5 GHz, since the beamformer chip was limited to operating only in the LMDS band. Each channel of the beamformer was calibrated as each cable’s curvature introduced different phases.

The gain of the phased array was referenced to the measurement of a horn antenna with a 20.8 dB gain connected to one channel of the beamformer. Table 2 shows the measured gain, the simulated directivity and the efficiency of the antenna at the central frequency of 26.5 GHz. The antenna offers a maximum realized gain of 15.2 dB in the broadside direction and a minimum at the maximum α = 140°. However, the efficiency is much higher in the latter case, peaking up to 91.2%.

Figure 10 shows the realized gain in the azimuth plane at the central frequency of 26.5 GHz, showing an average value of 1 dB less gain than in the simulated case, due to the manufacturing process. The maximum amount of sidelobe level (SLL) was measured to be 12 dB for a progressive phase of 140°, which can be reduced because a uniform distribution among the channels was used along the array. We can see from Figure 11 that there is also a good correlation of the gains with respect to frequency. Only three different progressive phases corresponding to α = 0°, 56° and 140° were plotted in order to keep the figure clearer.

## 5. Comparison with Other Work

Table 3 shows a comparison with other studies using 3D-printing technology for horn antennas. There are mainly two types of printing technology, classified as metal or plastic printing. In [12], a 16 × 16 horn array printed using DMLS is considered. Although the complexity is high, it uses metal printing, which increases the fabrication costs. The manufacturing cost and the weight can be considerably reduced by using plastic printing technologies such as the popular FDM. In [13,14,15], the authors all use the same FDM technique, with copper electrodeposition for the metallization of the whole of the inner and outer plastic parts. However, as the frequency increases and the manufacturing complexity increases, there will be tiny parts that can be very difficult to metallize. In this study, a much simpler metallization procedure was used, so that only the outer part of the horn needed to be metallized, as stated in Section 3.1.

In addition, for many upcoming future 5G applications, interconnection between the radiating elements and RF chipsets is required for effective integration with the antenna system. This aspect is reported in this work but not provided in any other 3D-printed horn antenna studies. The gain of the horn antenna is lower than in most of the comparison studies because the effective aperture is much smaller and the array consisted of only four active H-sectorial horn elements, but it could be increased by adding more active elements or by enlarging the aperture in the H-plane.

## 6. Conclusions

A 3D-printed phased array of four H-sectorial horns that minimized the blind scanning problem and provided an easier metallization solution was proposed. A microstrip-line-to-rectangular-waveguide transition was designed for the integration of the phased array with a commercial digital beamformer. The transition offered a simulated FBW of 22.5%, with a maximum single-transition IL of up to 0.67 dB and a minimum IL of 0.3 dB. The 3D-printed phased array was used to scan from 0° to 140° at the central frequency of 26.5 GHz in steps of 28°, giving a total of six patterns. The measured gain of the array was 15.2 dB for the case of the broadside direction, with a measured FBW of 20.8% for the case of a single-port measurement. The results proved the feasibility of 3D-printing technology for the manufacturing of phased arrays in high-frequency bands and their easy integration with digital beamformers. A future improvement would be to integrate the antenna directly with beamforming chips to avoid the use of coaxial interconnections, which introduce additional losses.

The antenna is suitable for the whole LMDS and K-band for 5G millimeter-wave applications requiring low cost, high gain and integrated solutions.

## Figures and Tables

**Figure 1 sensors-22-05329-f001:**
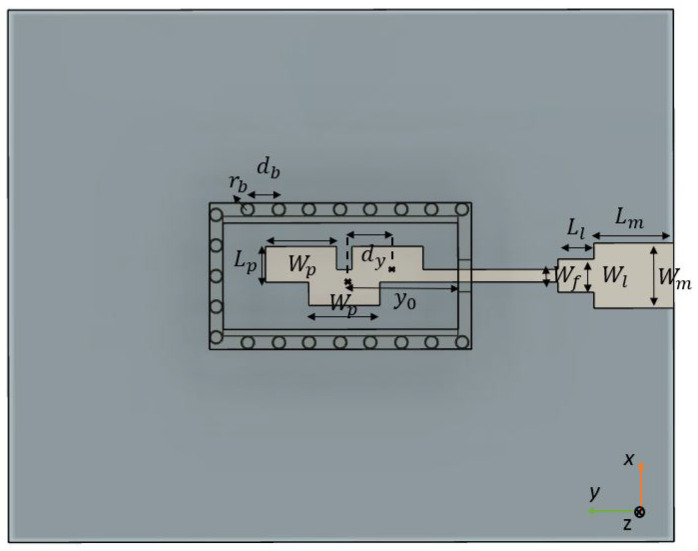
Top view of the transition.

**Figure 2 sensors-22-05329-f002:**
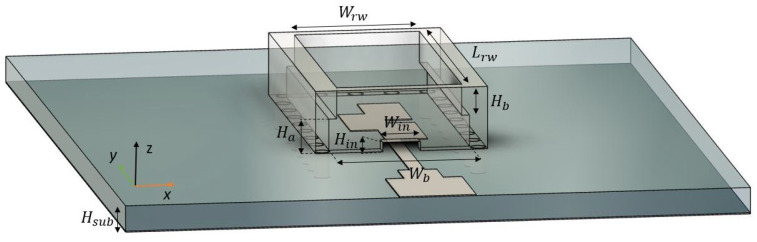
Isometric view of the transition.

**Figure 3 sensors-22-05329-f003:**
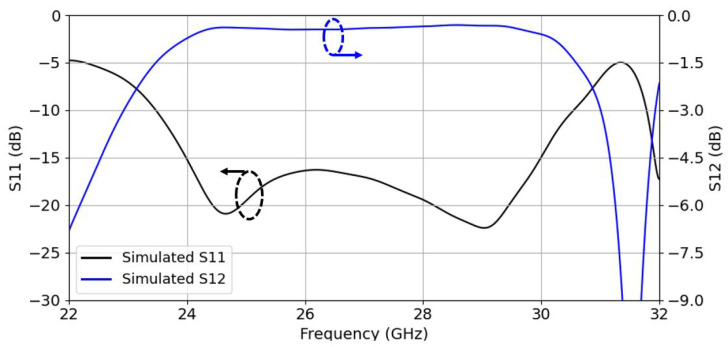
S-parameters of the single transition.

**Figure 4 sensors-22-05329-f004:**
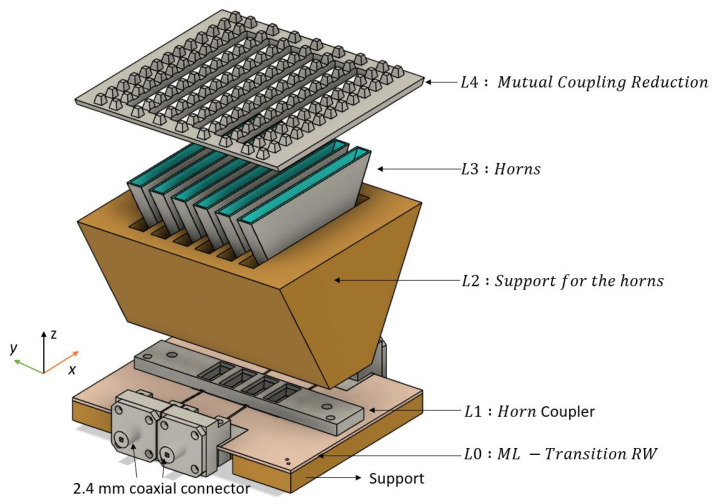
Exploded view of the whole design.

**Figure 5 sensors-22-05329-f005:**
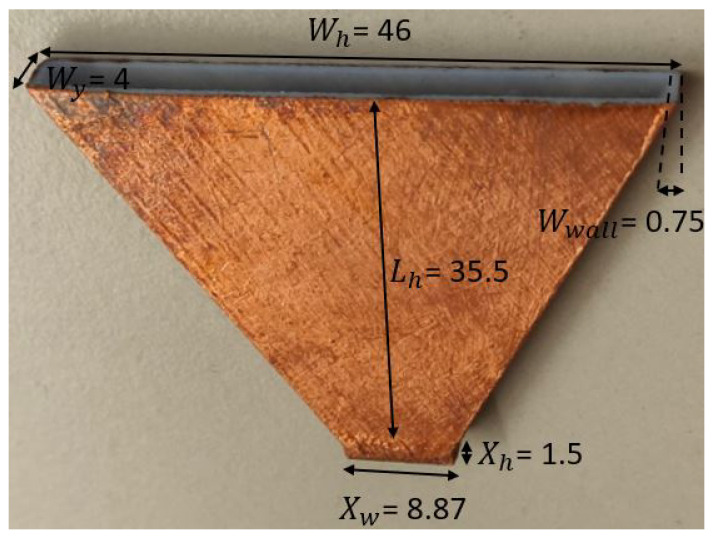
Horn antenna (all dimensions in mm).

**Figure 6 sensors-22-05329-f006:**
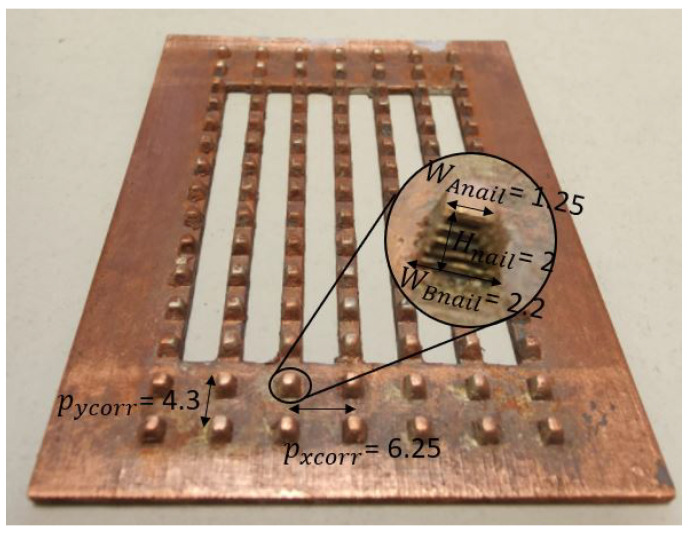
L4 layer showing the corrugations used for blind scanning reduction of a phased array.

**Figure 7 sensors-22-05329-f007:**
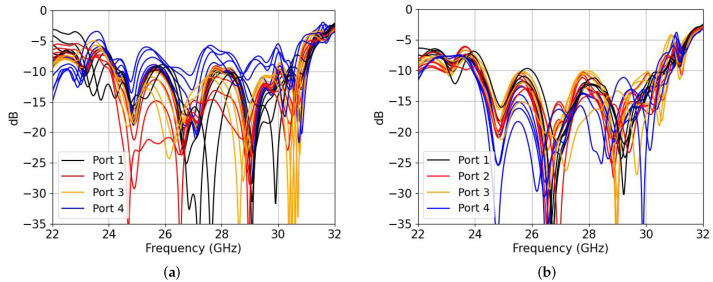
Simulated input reflection coefficients for each port: (**a**) without nails; (**b**) with nails. There are 6 traces per port relating to the progressive phase from 0° to 140° at 26.5 GHz.

**Figure 8 sensors-22-05329-f008:**
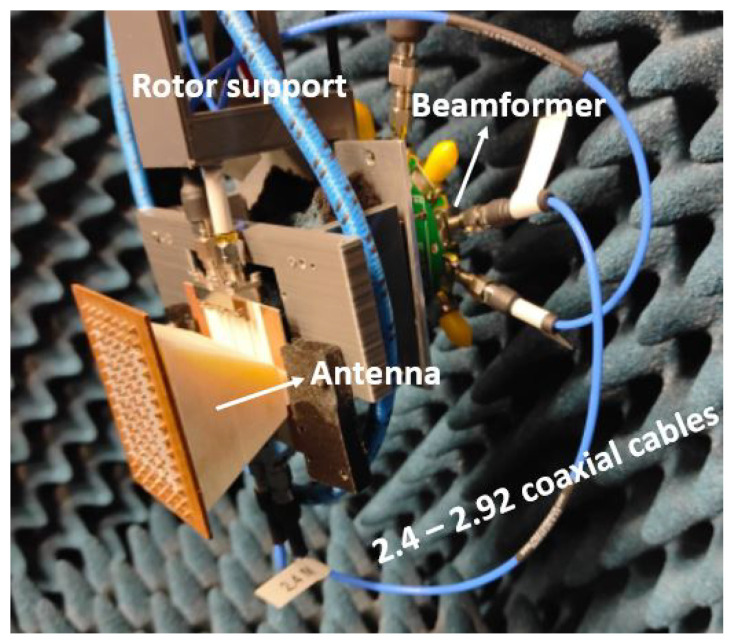
Entire manufactured phased array connected with the beamformer and ready to be measured in the anechoic chamber.

**Figure 9 sensors-22-05329-f009:**
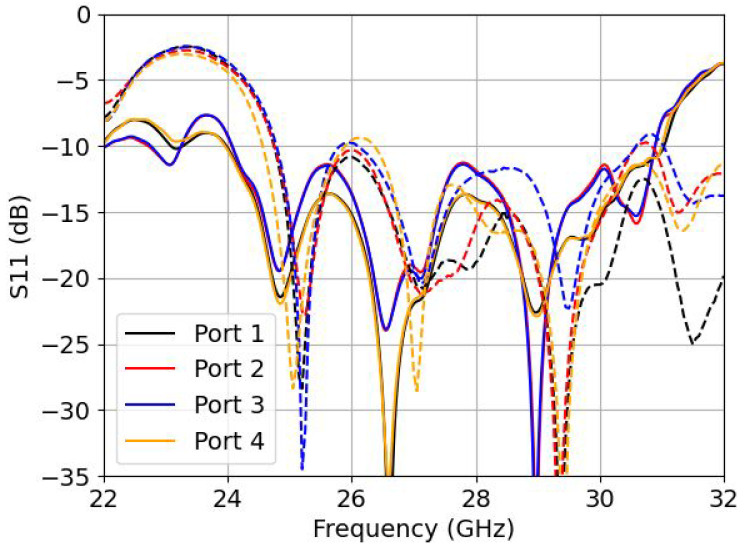
Input reflection coefficient (for measured - - and simulated -) for each port independently.

**Figure 10 sensors-22-05329-f010:**
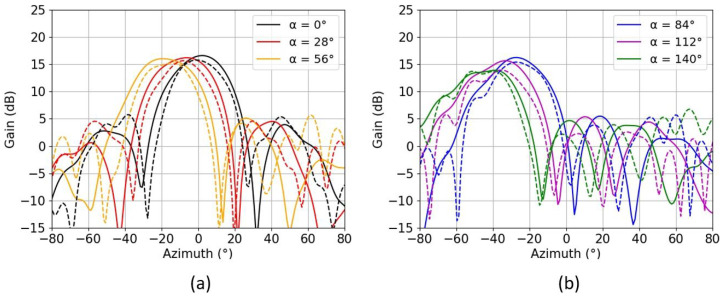
Radiation pattern in the E-plane (for measured - - and simulated -) for six different progressive phases α ranging from 0° to 56° (**a**) and from 84° to 140 ° (**b**), at 26.5 GHz.

**Figure 11 sensors-22-05329-f011:**
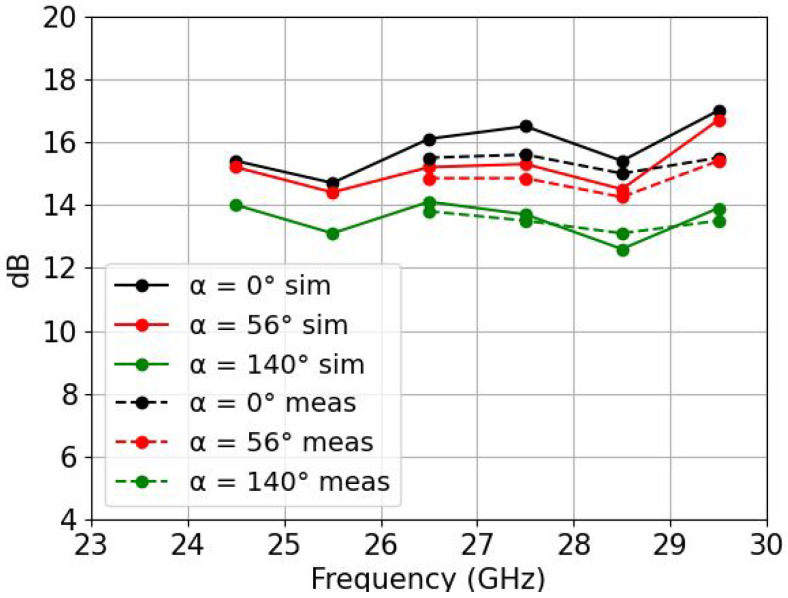
Realized gain over frequency for three different progressive phases, α = 0°, 56° and 140°. Values were measured only from 26.5 GHz to 29.5 GHz due to the limitations of the beamformer.

**Table 1 sensors-22-05329-t001:** Dimensions in mm of each designed parameter.

$H_{sub}$	0.81	$W_{f}$	0.47	$W_{rw}$	4.2	$r_{b}$	0.25
dy	1.5	$W_{l}$	1.1	$L_{rw}$	8.8	$d_{b}$	1.17
$W_{p}$	2.75	$L_{l}$	1.35	$H_{b}$	1	$H_{in}$	0.5
$L_{p}$	1.3	$L_{m}$	2.5	$H_{a}$	2	$W_{in}$	1.2
$Y_{0}$	4.48	$W_{m}$	2.55	$W_{b}$	4.5		

**Table 2 sensors-22-05329-t002:** Measured realized gain, simulated directivity in dB and efficiency η in % for the six different progressive phases α ranging from 0° to 140°.

α	0°	28°	56°	84°	112°	140°
Gain (dB)	15.2	15.15	15.04	15.05	13.75	13.7
Directivity (dB)	16.25	15.8	15.8	15.74	15.33	14.2
η (%)	78.5	86	83.9	85.3	69.5	91.2

**Table 3 sensors-22-05329-t003:** Comparison with other studies using 3D printing technology for horn antennas.

Ref.	Gain (dB)	Aperture Size (λ)	Elements	Printing Technique	Bandwidth (GHz)	Chip Integration
[12]	33.8	13 × 13	256	DMLS	28.2–35.8	No
[13]	13.51	1.15 × 1.15	1	FDM	3.5–11.4	No
[14]	18.7	4 × 4	1	FDM	8.2–12.4	No
[15]	20	2.7 × 3.68	1	FDM	25–40	No
This work	15.2	2 × 3.9	4	MJ	24.75–30.3	Yes

## Data Availability

Not applicable.

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
