# Peer review of "Microstrip-Fed 3D-Printed H-Sectorial Horn Phased Array"

_sensors, 2022, doi:10.3390/s22145329_

Round 1

Reviewer 1 Report

The manuscript presents the study performed for 3D printed horn phase array feed mechanism. Below are my comments.

Abstract: What are the values of directivity and how much is the difference value between directivity and gain values. What is your design frequency? What is the weight of your proposed antenna?

1.       Introduction

The use of 3D printed system needs to more researched. Its very few. Do some more study.

2.       Transition Design

What is see Fig. ??

What is the distance between 3D printed PA and feed source?

In figure3, the label should not overlap your graph!!! Modify the figure

3.       Phased Array Design

Its ok

3.1   Horn Antenna

Why inner layer of horn is not metallized?? Need more explanation

3.2   Mutual Coupling Reduction

Figure 7 (a) and (b) seems to be mesh. What is the meaning of accommodating all graphs in one. Explain it more.

4.       Results

Figure 10 is very mesh. Have separate plots with minimal alpha values.

For steered angle case, what is the elevation angle in your case??

Why your gain is very less for alpha 140-degree case. How much is the variation among the gain values for different alpha values??

Put the comparison table against the other proposed literature antenna prototypes. Focus on 3D printed antennas.

Plot the VSWR curves. Is your proposed antenna a wideband type in nature?? Do justify it.

5.       Conclusion

Its ok

References

Add more references. Its too few. Do some more study.

Do provide line by line responses with your comments for the revised manuscript. 

Author Response

Dear reviewer, 

Please find the response to your review in doc format. Thank you really much for your time dedicated on reviewing this paper. 

Regards,

Ivan Zhou

Reviewer 2 Report

A 3D printed phased array consisting in four H-Sectorial horn antennas was designed and VNA measured. The paper is well written. Some comments are as follows:

1, In the Abstracts, what's the meaning of "LMDS"?

2, Third paragraph of Introduction, "transitions from the horn (RW)", The 'RW' is short for "transitions from the horn"??

3, In the Introduction, please explain the novelty of this proposed antenna obviously.

4, In Section 2, "TE01 mode", Please check it is TE_0_1 or TE_1_0 for the rectangular waveguide?

5, In the CST simulation, please give the substrate's dielectric permittivity and loss tangent.

6, Irregular writings in the text, such as TE01, S11 and S21 should be TE01S11 and S21. Below Fig.3, "The narrow wall of the RWis stepped (see Fig. ??)".

7, Please explain the 3D printing tolerance in the paper.

8, Lines 134 and 135 should not be separated but one sentence.

Author Response

Dear reviewer, 

Please find attached the revision document in word format. 

Thank you for your help ! 

Regards,

Ivan Zhou

Round 2

Reviewer 1 Report

The manuscript is well revised. Just a few minor correction is need with Figure 11. There is the overlap of curves with label description. Do modify it. 

Author Response

Dear reviewer,

I have updated the figure 11, here you can find the corrected version in pdf.

Thank you very much for all your time spent for improving our paper !

Regards,

Ivan Zhou 
